# Fungal Drug Response and Antimicrobial Resistance

**DOI:** 10.3390/jof9050565

**Published:** 2023-05-12

**Authors:** Paloma Osset-Trénor, Amparo Pascual-Ahuir, Markus Proft

**Affiliations:** 1Department of Biotechnology, Instituto de Biología Molecular y Celular de Plantas IBMCP, Universidad Politécnica de Valencia, 46022 Valencia, Spain; 2Department of Molecular and Cellular Pathology and Therapy, Instituto de Biomedicina de Valencia IBV-CSIC, Consejo Superior de Investigaciones Científicas CSIC, 46010 Valencia, Spain

**Keywords:** antifungal resistance, pleiotropic drug response, pathogenic fungi, multidrug efflux, antifungal drugs

## Abstract

Antifungal resistance is a growing concern as it poses a significant threat to public health. Fungal infections are a significant cause of morbidity and mortality, especially in immunocompromised individuals. The limited number of antifungal agents and the emergence of resistance have led to a critical need to understand the mechanisms of antifungal drug resistance. This review provides an overview of the importance of antifungal resistance, the classes of antifungal agents, and their mode of action. It highlights the molecular mechanisms of antifungal drug resistance, including alterations in drug modification, activation, and availability. In addition, the review discusses the response to drugs via the regulation of multidrug efflux systems and antifungal drug–target interactions. We emphasize the importance of understanding the molecular mechanisms of antifungal drug resistance to develop effective strategies to combat the emergence of resistance and highlight the need for continued research to identify new targets for antifungal drug development and explore alternative therapeutic options to overcome resistance. Overall, an understanding of antifungal drug resistance and its mechanisms will be indispensable for the field of antifungal drug development and clinical management of fungal infections.

## 1. Introduction

Fungi can cause a variety of human diseases [1], including (i) local mycoses of the skin, nails, and hair, (ii) systemic mycoses that spread throughout the body and affect internal organs, such as histoplasmosis, aspergillosis, and candidiasis, (iii) opportunistic infections that occur in people with weakened immune systems, such as *Pneumocystis jirovecii* and *Cryptococcus neoformans*, (iv) allergic diseases triggered by an allergic reaction to fungal spores such as bronchopulmonary aspergillosis, or (v) toxigenic infections caused by toxic substances produced by fungi, such as aflatoxicosis and ergotism. Every year, over 150 million severe fungal infections occur globally, leading to around 1.7 million deaths annually. Disturbingly, these numbers are steadily increasing due to social and medical advancements that have facilitated the spread of these infections. Furthermore, the long-term use of antifungal drugs in high-risk patients has resulted in the emergence of drug-resistant fungi, including the highly dangerous *Candida auris* [2,3]. Very recently, the World Health Organization (WHO) has raised the alert of threatening fungal infections by ranking a total of 19 fungal species as priority pathogens for strategic research, development, and public health actions [4]. The most critical group contains *Cryptococcus neoformans*, *Candida auris*, *Aspergillus fumigatus*, and *Candida albicans.* According to this ultimate analysis from WHO, research into fungal-invasive diseases still receives insufficient funding, with only 1.5% of the total research budget in infectious diseases, and the approved antifungals and those in clinical trials do not fully solve the challenges posed to healthcare workers due to treatment limitations and increasing resistance. The widespread use of fungicides (azoles) in agriculture additionally exacerbates the issue, thus demanding the development of new types of antifungals with different targets, safer profiles, and mechanisms of action exclusively for human use. Thus, fungal infections are a growing global concern that requires more effective research to understand and combat fungal antimicrobial resistance [3,5]. This article summarizes and highlights the molecular mechanisms underlying the adaptive response and acquired resistance to different classes of antifungal compounds.

## 2. Antifungal Drugs: Classes and Modes of Action

Currently, only four classes of systemic antifungal treatments are used in clinical practice: azoles, echinocandins, polyenes, and pyrimidines [6] (Figure 1). Azole antifungal drugs, first reported in the late 1960s [7], work by inhibiting the synthesis of ergosterol, an essential component of fungal cell membranes. This results in increased permeability of the fungal cell, leading to its destruction. Azoles are used to treat a variety of fungal infections, including dermatophytosis, candidiasis, and aspergillosis. There are several classes of azole antifungal drugs [8], including (i) imidazoles, such as clotrimazole, econazole, and miconazole, which are commonly used to treat skin and nail infections, (ii) triazoles, such as fluconazole, itraconazole, and voriconazole, which are used to treat systemic fungal infections and (iii) allylamines, such as terbinafine, which is used to treat dermatophyte infections. Ergosterol acts as a control mechanism for the fluidity, asymmetry, and overall stability of the cell membrane in fungal cells [9]. For the cell membrane to maintain its integrity, the incorporated sterols must not have a C-4 methyl group. Azoles primarily impact the heme protein Cyp51 (Erg11), which catalyzes the cytochrome P-450-dependent removal of a methyl group from lanosterol through the process of 14α-demethylation within the ergosterol biosynthesis pathway [10,11]. Azole drugs inhibit the enzyme reaction by non-competitive reversible interaction with the heme and prevent accessing of protons to the active site [12]. When the 14α-demethylase is inhibited, the levels of ergosterol decrease, and the levels of its precursors, such as lanosterol, 4,14-dimethylzymosterol, and 24-methylenedihydrolanosterol, increase, leading to structural and functional changes in the plasma membrane [13].

Echinocandins are a class of antifungal agents discovered in the 1980s and 1990s. They work by targeting a key component of the fungal cell wall, β(1,3)-glucan, which is essential for its stability and integrity (Figure 1). The unique composition of the fungal cell wall, which includes compounds such as mannan, chitin, and α- and β-glucans, has made it a prime target for antifungal agents [14]. These components are exclusive to fungi and not present in other organisms, making them a desirable target for drugs to selectively target and kill fungal cells while minimizing damage to other cells [15]. The majority of our understanding of the cell wall composition of fungi that are medically significant comes from research conducted on *C. albicans* [16]. This yeast species has a complex cell wall structure made up of multiple layers, including chitin, β-glucan, and mannoprotein. These last two components make up approximately 80% of the overall mass of the cell wall, thus highlighting the importance of β-glucan for fungal cell stability [17]. Echinocandins inhibit the synthesis of β(1,3)-glucan by binding to the enzyme β(1,3)-glucan synthase, a large (210 kDa) integral membrane heterodimeric protein, thereby disrupting the fungal cell wall and leading to cell death [18,19]. The mechanism of action of echinocandins makes them effective against a variety of clinically important fungi, including *Candida* and *Aspergillus* species. They are typically used to treat serious invasive fungal infections, often in combination with other antifungals.

Polyenes are a class of antifungal agents discovered in the 1950s [20]. Similar to azoles, they target the unique fungal membrane component ergosterol; however, in this case, by direct binding to the sterol, thereby causing a change in the permeability of the membrane and disruption of the membrane potential [21,22] (Figure 1). Polyenes have a broad spectrum of activity against various medically important fungi, as they inactivate all cells with a significant content of sterols in their outer membranes [23]. The most important drawback of the use of polyenes is their intrinsically high host toxicity, which limits their therapeutic index [24,25]. One of the most well-known polyenes is amphotericin B, which was initially derived from the soil bacterium *Streptomyces nodosus*. Amphotericin B was the first effective antifungal agent and is still widely used today, together with nystatin and natamycin, especially for the treatment of serious or life-threatening fungal infections [20,26].

Pyrimidine analogs are a different class of antifungal agents that act by inhibiting DNA and RNA synthesis [27]. Flucytosine (5FC) is the most commonly used pyrimidine analog for the treatment of fungal infections. The 5FC has been shown to inhibit the growth of various yeasts, including *Candida* and *Cryptococcus neoformans*. However, the high prevalence of primary resistance in many fungal species has reduced its initial promise as an antifungal agent [28,29,30]. Nowadays, 5FC is mostly used in combination with other antifungals, such as amphotericin B and fluconazole, and is rarely used as a single agent. The mechanism of action of 5FC involves entering the fungal cell through a permease enzyme and being converted to 5-fluorouracil (5FU) by the enzyme cytosine deaminase. This is followed by the conversion of 5FU into 5-fluorouridylic acid (FUMP) by UMP pyrophosphorylase, which is further phosphorylated and incorporated into RNA, causing disruption of protein synthesis [31]. Additionally, 5FU is converted to 5-fluorodeoxyuridine monophosphate, a potent inhibitor of thymidylate synthase, an enzyme involved in DNA synthesis and nuclear division [32]. Thus, 5FC acts by disrupting pyrimidine metabolism, as well as RNA, DNA, and protein synthesis in the fungal cell (Figure 1).

Although there is a growing need for more antifungal drug options [33,34], no new classes of antifungal drugs have become available in the past 20 years. However, there is some good news, as several new antifungal classes are currently in late-stage clinical development. These include promising drugs currently in clinical trials with previously unexplored modes of action, such as fosmanogepix (a novel inhibitor of the Gwt1 enzyme) or olorofim (a new inhibitor of the dihydroorotate dehydrogenase enzyme) [35,36,37] (Figure 1).

Fosfomanogepix is a N-phosphonooxymethyl prodrug, which after drug administration, is converted into the active form of manogepix by systemic phosphatases [38]. Manogepix has a new mode of inhibition that targets the maturation of glycosylphosphatidylinositol (GPI)-anchored proteins by inhibiting the fungal enzyme Gwt1 at the endoplasmic reticulum [39]. This enzyme, which is crucial for the trafficking and anchoring of mannoproteins to the fungal cell membrane and wall, is an inositol acyltransferase [40]. GPI-anchored mannoproteins act as adhesives, enabling fungi to stick to mucosal and epithelial surfaces in the host before colonization and infection. Several fungal adhesins and virulence factors are derived from GPI-anchored proteins and GPI biosynthesis is essential in fungi [41,42]. The inhibition by manogepix appears to be unique to fungal pathogens because it does not impede human inositol acylation via the nearest mammalian ortholog, PIGW [38,43]. Manogepix has broad-spectrum activity against numerous pathogenic fungi including *Candida* and *Aspergillus* spp. [44,45].

Olorofim is a member of a novel class of antifungal drugs named orotomides. Olorofim impairs fungal growth through inhibition of the fungal dihydroorotate dehydrogenase (DHODH) enzyme, which is located in mitochondria and involved in pyrimidine synthesis [46,47] (Figure 1). Olorofim does not show any significant cross-reactivity with the human dihydroorotate dehydrogenase, thus limiting its drug toxicity [48]. While olorofim may not be effective against a wide range of fungal infections, it possesses remarkable novelty and a spectrum of activity that is anticipated to be significant in the future. Olorofim does not demonstrate any efficacy against yeast. Nevertheless, it is effective against various clinically important fungal groups, including dimorphic molds such as *Histoplasma* and *Coccidioides* spp. or hyaline hyphomycetes such as *Aspergillus* spp. [35,46].

## 3. Impact of Fungal Antimicrobial Resistance

Infections are a leading cause of mortality and disability worldwide, with drug-resistant bacterial infections causing 1.27 million direct deaths and contributing to an estimated 4.95 million deaths annually [49]. This burden is particularly high in resource-limited settings. Fungal diseases cause over 1.5 million deaths and affect over a billion individuals [50]. Serious fungal infections often occur as a consequence of underlying health conditions such as asthma, AIDS, cancer, organ transplantation, and corticosteroid therapies. The WHO recognizes that the incidence of invasive fungal diseases (IFDs) is increasing globally, particularly among immunocompromised populations [4]. However, there are challenges in diagnosing and treating IFDs, including limited access to quality diagnostics and treatments and the emergence of antifungal resistance. Despite the growing concern, fungal infections receive limited attention and resources, resulting in a scarcity of data on their distribution and patterns of antifungal resistance [4,51,52,53]. Thus, it is not possible to accurately estimate their burden.

The acquisition and emergence of antifungal drug resistance can be explained as an evolutionary response to the selective pressure exerted by the drug. The likelihood of resistance emerging due to genetic changes is influenced by several factors, including the number and doubling rate of fungal cells exposed to the drug, the number of different pathways that confer resistance, and the fitness costs associated with each pathway [54]. It is important to note that antifungal drug resistance can originate either in the host or in the environment. In vivo resistance develops in individuals during antifungal therapy, leading to treatment failure for a range of pathogenic fungi [55,56]. This is particularly relevant for *Candida* spp., which is a major cause of nosocomial bloodstream infections and frequently shows the emergence of resistance to antifungals [57]. For example, azole resistance in *C. albicans* has been documented in individuals with HIV who were undergoing prolonged fluconazole therapy [58]. The demographics of patients susceptible to IFDs are progressively broadening, with a particular emphasis on populations such as the elderly, individuals with compromised immune systems due to HIV, cancer chemotherapy, or immune-suppression therapy required for transplantation, as well as those suffering from severe viral infections such as influenza virus and COVID-19 [59,60,61,62]. Environmental resistance can also emerge due to prior exposure of human pathogenic fungi to fungicides in nature. Fungicides are commonly used to protect crops and materials against fungal infections, and the resulting selective pressure drives the evolution of resistance against all major classes of fungicides, including azoles [63].

## 4. Mechanisms of Acquired Antifungal Resistance

At a mechanistic level, antifungal resistance is typically acquired through modifications that impact the interaction between the drug and its target, either directly or indirectly [64,65]. The development of resistance can be attributed to genetic alterations in the binding site of the target, such as mutations in the genes encoding lanosterol demethylase for azoles or β-glucan synthase for echinocandins [6]. Resistance may also result from an increase in the amount of target available [66], or through changes in the effective drug concentration, as a consequence of elevated drug efflux activity for intracellular drugs such as azoles [67,68], or through the inhibition of prodrug activation for flucytosine [69]. In scientific terminology, antifungal resistance is distinct from antifungal tolerance, which refers to the capacity of drug-sensitive cells to survive in the presence of drug concentrations beyond the minimum inhibitory concentration (MIC). This ability is associated with various general stress responses and/or epigenetic pathways, as comprehensively reviewed in [70]. Notably, tolerance is most pronounced when using fungistatic drugs and has been extensively examined in *Candida albicans* strains treated with fluconazole [71]. Nevertheless, the clinical significance of antifungal tolerance remains uncertain. In this review, we focus on the molecular mechanisms, which enable fungal cells to respond and protect from antibiotic drugs and eventually develop drug resistance.

### 4.1. Fungal Multidrug Efflux Systems: Components, Regulation, and Impact on Drug Resistance

A notable proportion of observed clinical resistance to antifungal agents can be attributed to the action of ATP-binding cassette (ABC) and major facilitator superfamily (MFS) transport pumps located at the fungal plasma membrane [72,73]. The majority of studies in this field have been conducted with azole drugs [74]; however, multidrug efflux affects resistance to other classes of antifungals [75]. The ABC and MFS transporters are the two most extensively researched transporter families implicated in efflux [73]. Fungi allocate a significant portion of their genomic resources to encoding transporters, with fungal genomes containing approximately 10 to 30 transporter-encoding genes per megabase of genomic DNA [76]. The ABC transporter superfamily comprises primary efflux transporters, which use ATP hydrolysis to export substrates [77,78]. This superfamily can be further divided into five families of transporters (ABCA, ABCB, ABCC, ABCD, and ABCG), with three families being involved in the efflux of toxic compounds. These three families, known as multi-drug resistance (MDR), multi-drug resistance-associated protein (MRP), and pleiotropic drug resistance (PDR), respectively, have been extensively investigated in *Saccharomyces cerevisiae*, offering insights into their potential roles in pathogenic fungi [79,80]. The PDR family exhibits the lowest level of phylogenetic conservation among the ABC transporter families, with evidence of gene loss and duplication observed in yeasts and filamentous fungi [81]. This implies that the members of the PDR family evolve rapidly in response to external selective pressures.

The conserved structure of ABC transporters comprises a nucleotide-binding domain (NBD) located either preceding or following a transmembrane domain (TMD), which is formed by six transmembrane-spanning helices. This NBD-TMD_6_ configuration is observed in half ABC transporters, which dimerize to produce a complete functional protein. In contrast, most fungal ABC transporters have undergone evolutionary changes that have led to the fusion of two NBD-TMD_6_ half transporters, resulting in the formation of a single functional protein instead of a dimeric structure. ABC transporters are a highly complex and fascinating class of proteins. Although some members of this superfamily exhibit high substrate specificity, ABC multidrug transporters are highly promiscuous in their substrate recognition. These proteins harness the energy generated by ATP binding and hydrolysis at the NBDs to induce conformational changes that enable the transporters to bind and release drug substrates [82]. The binding pocket is constructed from the TMDs and can accommodate large substrates. The transporter adopts a high-affinity inward-facing (IF) conformation during the binding phase, while the substrate is released from the outward-facing (OF) structure with lower affinity. As a result, these proteins exhibit polytopic characteristics. The nature of the drug-binding and transport sites is relevant to understanding resistance mechanisms mediated by these transporters. For example, the large drug-binding pockets of P-glycoprotein and Pdr5 contain multiple overlapping sites [83,84]. Notably, it has been observed that a single drug can bind to multiple locations within these sites [85] and that different drugs can simultaneously bind to different locations within the binding pocket [86].

The clinical relevance of efflux pumps extends beyond ABC multidrug transporters, as some members of the MFS family also mediate hyper resistance to drugs [87]. MFS proteins are typically composed of 12 or 14 transmembrane helices, with each transmembrane domain consisting of 6 or 7 helix bundles that pack together to create a cavity lined with amino acids that bind transport substrates. These transporters are generally smaller than ABC transporters and, similar to their counterparts, exhibit considerable substrate promiscuity. However, MFS transporters utilize the energy derived from a proton motive force rather than ATP hydrolysis to facilitate efflux.

Extensive research has been conducted on fungal multidrug transporters, with a particular focus on those found in *Saccharomyces cerevisiae*, *Candida albicans*, and *Candida glabrata* [77,88,89,90]. A group of *S. cerevisiae* strains was discovered that exhibited resistance to a wide range of drugs and natural toxins, known as pleiotropic drug resistance (PDR) [91]. Subsequent studies identified the primary genes involved in PDR of budding yeast to be ABC and MFS transporters and, importantly, zinc cluster transcription factors (TFs) [92,93,94,95] (Figure 2). The best-characterized PDR ABC transporters, Pdr5, Snq2, and Yor1, each play a role in resistance against hundreds of functionally and structurally unrelated drugs [96]. The most frequently mutated loci in PDR strains were found to encode two paralogous zinc cluster TFs, Pdr1 and Pdr3, which share 36% identity and are an essential part of the PDR network [97]. Recently, it has been shown in yeast that multiple PDR TFs contribute to the overall multidrug response with distinguishable sensitivities towards different xenobiotic molecules [98]. Pdr TFs were found to bind to pleiotropic drug resistance elements (PDREs) within the promoters of different multidrug transporter encoding genes, which is essential for PDR inducibility [99,100,101,102]. Pdr1 and Pdr3 are constitutively bound to DNA, regardless of the presence of drugs, suggesting that their activation is not dependent on changes in nuclear occupancy or DNA-binding [103]. Thakur et al. [104] showed that the regulatory domains XBD (xenobiotic binding domains) of both Pdr1 and Pdr3 bind the antifungal ketoconazole at low micromolar affinity, suggesting that ligand binding may increase their transcriptional activity. Similar to other known zinc cluster proteins, partial deletions or mutations within the regulatory domains of Pdr1 and/or Pdr3 result in constitutively active TFs that induce strong expression of ABC transporters [105,106,107], leading to an instance of up to 10-fold higher minimum inhibitory concentrations of ketoconazole. The pathogenic fungi *Candida glabrata* and *Candida albicans* have different mechanisms for controlling the expression of their drug efflux ABC transporters, with Pdr1 regulating Cdr1, Pdh1, and Snq2 transporters in *C. glabrata* [108,109] and Tac1 and Mrr1 controlling the expression of Cdr1, Cdr2 or Mdr1 transporters in *C. albicans* [110,111,112,113]. *Aspergillus fumigatus* lacks a Pdr1 homolog but has intrinsic drug resistance through AbcG1 and its zinc cluster TF regulator AtrR [114,115].

The major mechanism of acquired antifungal resistance via drug efflux is the reinforcement of the expression of one or several efflux transporters (Figure 2). This can be achieved by an increase of genomic copies of the relevant *PDR* genes (gene amplification), the gain of function mutations in PDR TFs, or modification of PDR TFs mRNA stability. Amplification of genes, including *ERG11* and *TAC1*, plays a significant role in the multidrug resistance of *Candida* species [116,117,118], leading to a linear relationship between gene copy number and the minimum inhibitory concentration of fluconazole. Aneuploidy leading to drug resistance has been observed in *C. albicans*, *C. neoformans*, and *C. auris* [119,120,121]. These findings show that increasing the copy number of genes involved in multidrug efflux, even modestly, creates enough hyper-resistance to have clinical implications. More recently, the fast and reversible amplification of *PDR* genes leading to drug resistance in *C. albicans* has been investigated mechanistically, revealing the exchange between long, inverted repeats flanking the amplified region on multiple chromosomes. It is evident from these and other results that hyper resistance by gene duplication is frequent in many fungal species and can be achieved by multiple mechanisms, including aneuploidy, nonhomologous extension after double-strand breaks, and chromosomal exchange between indirect and direct repeats [122]. Because aneuploidities generally reduce fitness, they seem to arise only during severe changes in environment such as drug exposure, and do not persist once the selective pressure is removed [123,124]. Remarkably, aneuploidities were found in a great portion of clinical *S. cerevisiae* strains [123].

Pleiotropic drug resistance by gain of function (GOF) mutations in TFs is dominated in budding yeast by the zinc cluster factors Pdr1 and Pdr3. It is important to note that, although not understood in detail on many occasions, these GOF mutations generally do not lead to overexpression of the TFs; instead, they create hyperactive TFs, which constitutively overexpress their transporter target genes. Amino acid substitutions in Pdr1, such as M308I in *PDR1-2*, F815S in *PDR1-3*, K302Q in *PDR1-6*, P298A in *PDR1-7* and L1036W in *PDR1-8* mutants, all result in multidrug-resistant phenotypes [105]. Three regions within the Pdr1 regulator with implications for drug resistance were identified, and mutant alleles *PDR1-3*, *PDR1-6*, and *PDR1-8* were constructed to investigate their phenotypes, with *PDR1-3* being the most effective in activating target promoters [105]. Similar to Pdr1, multiple amino acid substitutions in Pdr3, which cluster in short regions of the TF, can cause drug resistance [106]. According to [125], the presence of GOF mutations isolated from patients suggests that the *Candida glabrata* homolog Pdr1 plays an important role in the development of clinical antifungal resistance. In this case, Pdr1 expression itself is heavily regulated; thus, even a modest up-regulation of the wild-type Pdr1 copy contributes to drug resistance in this pathogenic fungus [126]. Extensive studies on CgPdr1 have shown an extraordinarily high number of GOF substitutions, which lead to enhanced drug resistance [125,127,128,129]. Recent advances point to the existence of an extensive regulatory domain within CgPdr1, which contributes to the highly flexible transactivation capacity via contacts with other co-activator proteins [126,130]. In *Candida albicans*, Tac1 is the main contributor to drug resistance via GOF mutations. Many points and small deletion mutations have been correlated in this TF with antifungal resistance by enhancing its transactivation capacity in the context with additional transcriptional co-activator complexes [112,131,132,133]. Additionally, multidrug resistance in *C. albicans* is caused by the Mrr1 regulator, which resembles Tac1. Mutations in Mrr1 can confer hyper resistance to benomyl, with many GOF mutations found in distinct locations exhibiting a dominant phenotype [113,134]. In the emerging high-risk pathogen *Candida auris*, two genes similar to *C. albicans’ TAC1* were identified, with a majority of GOF mutations reported in one of them, *TAC1B*, that map to regions analogous to those in the *C. albicans* homolog [135,136]. This suggests a potential link between the regulatory mechanisms of the two proteins. Rahman et al. [137] identified a novel mechanism of hyper resistance in *S. cerevisiae* involving a set of serine residues in the Pdr5 linker region. Their study showed that alanine mutations in six different serines exhibited significant hyper resistance, and changing S837 to aspartate produced the same phenotype, which was the result of a 2–3x increase of Pdr5 that was due to an increased Pdr5 mRNA half-life. Mixed effects on both PDR transporter expression and mRNA stabilization had been previously characterized in azole-resistant *C. albicans* isolates [138]. In any case, it is important to note that overexpression of multidrug efflux transporters interferes with general cell homeostasis, most likely by interfering with proper intracellular metabolites [139], thus only emerges upon a strong selection in the presence of antifungal agents.

A novel mechanism of antifungal resistance has emerged recently by investigating the A666G mutation in the yeast Pdr5 transporter, which was found in several screens for hyper-resistant mutants [140,141]. It conferred 2–5x increased substrate-specific resistance but did not alter susceptibility to clotrimazole. Importantly, the mutant had no difference in the quantity of Pdr5 or the level of ATPase activity compared to the wild type, but it had enhanced cooperativity, which led to higher transport of xenobiotic compounds per molecule of ATP hydrolyzed [82,141].

### 4.2. Mutational Inhibition of Antifungal Drug–Target Interaction

Several antifungal drugs (azoles, echinocandins, manogepix, or orotomides) directly target specific essential enzymes in the pathogen, such as Erg11/Cyp51, Fks1, Gwt1, or DHODH. Therefore, conformational changes in the target enzyme caused by mutations leading to decreased drug recognition while maintaining sufficient enzyme activity are possible sources of acquired resistance.

Mutations in the *ERG11(CYP51)* gene are a common azole resistance mechanism in *Candida albicans*, altering the inhibition of the lanosterol demethylase enzyme [142,143,144]. The effect of these mutations, more than 140 described to date, is not uniform and likely a result of overall reduced azole-binding affinity to the mutant enzyme [145,146,147]. Documentation of new *ERG11* mutations in azole-resistant clinical isolates continues to appear, but not all documented mutations were definitively shown to be tied to azole resistance [148,149,150,151,152,153]. Homozygous replacement of the native *ERG11* alleles with a mutant *ERG11* ORF encoding either a single or double amino acid substitution in the Erg11 protein showed that some substitutions directly influence or interfere with azole binding, while others indirectly affect resistance through, for example, the interaction with cytochrome P450 reductase [154]. The crystal structure of the Erg11 protein of *C. albicans* has been resolved, leading to further insight into the interactions between azole drugs and their target and the development of 3D models useful for future antifungal discovery [155,156]. *Aspergillus* spp. has two Cyp51 isoenzymes, Cyp51A, and Cyp51B, that can both act as sterol 14α-demethylase in vivo, resulting in no phenotypic difference in *A. fumigatus* strains with gene knock-outs of either enzyme [157,158,159]. However, a conditional mutant study showed that growth was abolished in the absence of both *CYP51A* and *CYP51B* [160]. The main azole resistance mechanism in *A. fumigatus* is the development of point mutations in *CYP51A*, which has less affinity for azole drugs than *CYP51B*, indicating a molecular basis for this resistance [161]. Lockhart et al. [162] reported that 60% of clinical *A. fumigatus* isolates had triazole resistance mutations within *CYP51A*, with the five most frequent mutations being G54, L98, G138, M220, and G448. These mutations have been shown to confer azole resistance in *A. fumigatus* through gene replacement experiments. Over 40 different *CYP51A* point mutations have been reported in clinical *A. fumigatus* isolates, but not all of them confer an azole resistance phenotype, and some can occur alone or in combination with other mutations [158,163,164,165,166,167,168,169]. Azole resistance is an increasing problem in *Cryptococcus* spp. due to prolonged treatment regimens, with *CYP51* mutations identified as one of the main mechanisms [170,171]. Point mutations in *CYP51*, such as G468S and Y145F, have been shown to confer resistance to fluconazole and voriconazole in *C. neoformans* [172,173].

The target of echinocandins, β(1,3)-glucan synthase, is encoded by the *FKS1* and *FKS2* genes in *Candida* spp. Limited amino acid substitutions in the Fks subunits of glucan synthase are responsible for echinocandin resistance leading to clinical failures [174]. These mutations occur in two highly conserved hot spot regions of *FKS1* in *C. albicans* [175] and in homologous regions of *FKS2* in *C. glabrata* [176], resulting in elevated minimum inhibitory concentration (MIC) values. The most frequent and pronounced resistance phenotype in *C. albicans* is caused by amino acid changes at S641 and S645, while in *C. glabrata*, modifications at S663 in Fks2, S629 in Fks1, and F659 in Fks2 are the most prominent substitutions [176,177]. Additionally, in *Aspergillus* spp., although much less explored, echinocandin resistance due to point-mutated Fks subunits has been reported recently [178].

Point mutations leading to amino acid substitutions within Gwt1, V163A in *C. glabrata*, and V162A in *C. albicans* can cause resistance to manogepix but do not affect the activity of other antifungals, including azoles and echinocandins [179,180]. The reduced susceptibility to manogepix of V162A and V168A Gwt1 mutants expressed in *C. albicans* and *Saccharomyces cerevisiae*, respectively, indicates the significance of this particular valine residue in manogepix binding across various species.

The olorofim target, dihydroorotate dehydrogenase (DHODH), encoded by the *pyrE* gene in *Aspergillus*, also appears to provide a limited resource for mutations leading to olorofim resistance. A recent study applying long-term olorofim treatment in *A. fumigatus* revealed several amino acid substitutions within PyrE with a hotspot at G119 producing olorofim resistance [181].

Polyene antifungals target an essential membrane compound, ergosterol, instead of an essential cellular enzyme. This different mechanism of action might explain the rare occurrence of polyene resistance among fungi. Nevertheless, the most prominent way to achieve polyene resistance is the alteration of the sterol composition of the plasma membrane [182,183], which comes with a strong fitness trade-off [184]. Several mutations in genes involved in the biosynthesis of ergosterol (*ERG* genes) have been linked to this mechanism. In *C. albicans*, when the *ERG11* and *ERG3* genes, which encode for lanosterol 14α-demethylase and C-5 sterol desaturase, respectively, are dysfunctional, the membrane’s ergosterol is replaced with alternate sterols such as lanosterol, eburicol, and 4,14-dimethyl-zymosterol [183,184]. According to other studies, the substitution in *ERG11* and the loss of function of *ERG5* (C-22 sterol desaturase) in *C. albicans* are associated with an alternative membrane sterol composition and amphotericin B resistance [150,184]. In other *Candida* species, the inactivation of *ERG6* (C-24 sterol methyl-transferase) and *ERG2* (C-8 sterol isomerase) has a similar effect [185,186]. A mutation in *ERG2*, which results in its inactivation, is one of the few described mechanisms of polyene resistance in *C. neoformans* [187].

### 4.3. Resistance by Overexpression of Antifungal Drug Targets

An important strategy of fungal cells to lower the efficient drug concentration inside the cell is based on titrating the antifungal compound through an increase in the copies of the target enzyme. A recent high-density mutagenesis screen in *S. cerevisiae* with several antimicrobial agents revealed that the gain of function promoter mutants is enriched for the respective drug target protein on many occasions, pointing to a general strategy to adapt to antifungal treatments [188]. Resistance by overexpression of the drug target is especially relevant for azoles. *ERG11* overexpression is a frequently observed mutational adaptation in clinically relevant azole-resistant *Candida* isolates [189,190,191,192]. One prominent mechanism of constitutive *ERG11* expression acts via the gain of function point mutations in the Upc2 TF, which acts as a direct transcriptional activator of *ERG11* in *Candida* spp. [193,194,195] In *Aspergillus*, different promoter rearrangements in the *cyp51A* gene leading to robust overexpression have been linked to azole resistance [196].

### 4.4. Alterations of Drug Modification, Activation, and Availability

Flucytosine (5-FC) stands out among antifungal agents due to its distinctive mode of action that involves inhibiting DNA, RNA, and protein synthesis and the fact that it acts as a prodrug, which means that it requires metabolic conversion through the pyrimidine salvage pathway to become activated [197]. The majority of research on 5-FC resistance mechanisms has concentrated on *S. cerevisiae* and *C. albicans*, with particular attention given to identifying mutations in crucial enzymes involved in the pyrimidine salvage pathways. These studies revealed that 5-FC resistance may occur due to the absence or alteration of any of the enzymes responsible for the activity of cytosine permease (*FCY2*), cytosine deaminase (*FCA1, FCY1*), or uracil phosphoribosyltransferase (*FUR1*). In *S. cerevisiae*, several studies have indicated that mutations in *FCY1*, *FCY2*, and *FUR1* can result in resistance to 5-FC [198,199]. Disruption of cytosine deaminase is linked to high levels of dose-independent resistance, whereas loss of cytosine permease results in a lower, dose-dependent level of resistance, suggesting the involvement of two additional cytosine permeases, *FCY21* and *FCY22*. Additionally, alternative entry routes of 5-FC exist via other cytosine adenine permease homologs, including *TPN1*, *FUR4*, and *YOR071c*, and can contribute significantly to 5-FC toxicity [200]. Kern et al. [201] have associated phenotypic 5-FC resistance with a single point mutation, R134S, in the *FUR1* gene. For *C. albicans*, acquired 5-FC resistance has been functionally linked to a single R101C amino acid substitution in the Fur1 enzyme [202,203], and several point mutations in the *FUR1* gene have been described in clinical 5-FC resistant isolates of *C. glabrata* [69]. In the emerging multiresistant *C. auris*, an I211F substitution in the *FUR1* gene has been documented in a 5-FC-resistant isolate [204]. Bhattacharya et al. [119] have further exhibited the potential of transient gene duplication in *C. auris* during replicative aging, leading to the development of tolerance to 5-FC.

Another related mechanism of resistance is the decrease of the effective drug concentration by the selective impairment of azole influx into the fungal cell. This has been recently reported to be relevant for PDR- and *ERG11*-independent azole-resistant *C. glabrata* isolates. Here, the molecular key to resistance is mutations in the CgHXT4/6/7 hexose importer, which contributes to azole import across the fungal cell membrane [205].

## 5. Concluding Remarks

The indiscriminate use of antifungal drugs for prophylactic and empirical treatment of suspected invasive fungal diseases in high-risk individuals such as patients with cystic fibrosis, critically ill, or haemato–oncology patients is a worrying trend. Antifungal stewardship is essential to ensure the appropriate use of these drugs and preserve the limited antifungal options available, particularly in cases of highly transmissible fungal infections such as *Candida* spp. and *Trichophyton* spp. [206]. In recent years, systems biology approaches utilizing laboratory evolution techniques have demonstrated the impressive ability of *Saccharomyces* and *Candida* species to acquire resistance to existing antifungal agents [207,208,209]. Despite the ongoing threat of drug-resistant fungal infections to human health, there is renewed optimism in academia and industry to enhance our knowledge of the mechanisms underlying antifungal resistance and to develop new therapeutic strategies to combat these pathogens. Combination antimicrobial treatment is an effective strategy to prevent secondary antimicrobial resistance and is established for various bacterial and viral infections. For example, combining antifungal agents such as amphotericin B and flucytosine or fluconazole and flucytosine can prevent the selection of resistant fungal populations and limit the development of clinical antifungal resistance [210,211]. Further investigation is required to unravel the full potential of combinatorial antifungal treatment in the battle against secondary resistance. Another promising approach is to target the resistance mechanisms themselves, with the aim of restoring the efficacy of current antifungals [212,213]. Alternative methods are currently being developed to protect antifungals from resistance, including host-directed approaches such as immunotherapy [214], fungal vaccines [215,216], and improved antifungal targeting [217]. In addition, cell-based therapies, such as dendritic cell transfer and CAR T-cell therapy [218,219,220,221], have demonstrated promising outcomes in vitro but still need to be investigated in clinical trials. These approaches provide potential alternative options for managing antifungal resistance. As such, ongoing and future investigations into the mechanisms of antifungal drug resistance are expected to yield important insights and pave the way for improved clinical outcomes for diverse populations affected by drug-resistant fungal infections worldwide.

## Figures and Tables

**Figure 1 jof-09-00565-f001:**
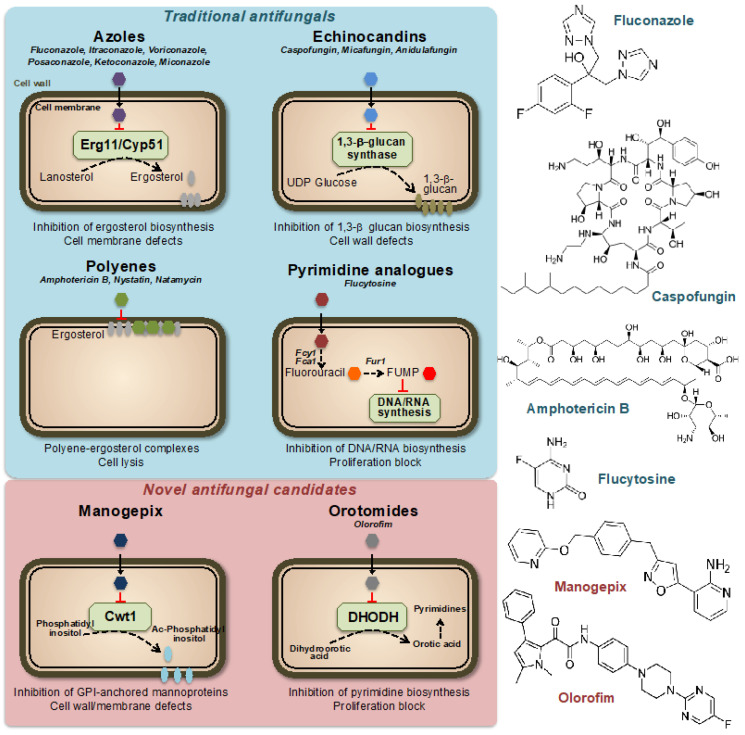
Mechanisms of action of traditional and novel antifungal drugs. Representative compounds are indicated for each class. FUMP = 5-fluorouridylic acid; Fcy1, Fca1 = cytosine deaminases; Fur1 = Uracil phosphoribosyltransferase; Ac-Phosphatidyl inositol = Acyl-Phosphatidyl inositol; GPI = Glycosylphosphatidyl inositol; DHODH = Dihydroorotate dehydrogenase.

**Figure 2 jof-09-00565-f002:**
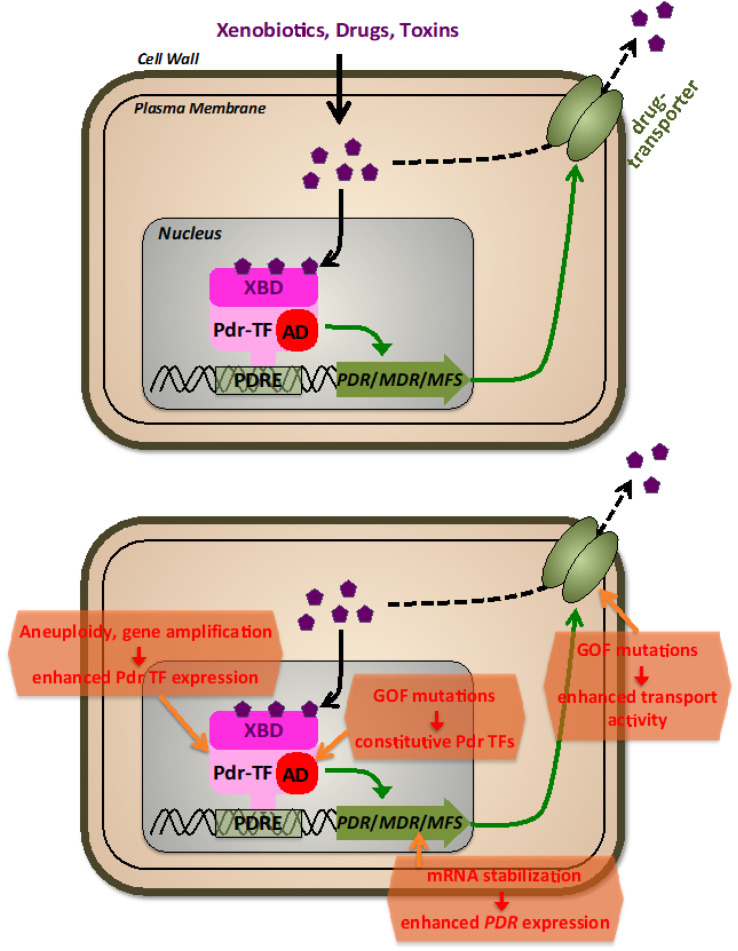
The fungal pleiotropic drug resistance (PDR). Upper panel: General features of PDR regulation. Different xenobiotic molecules, including antibiotic drugs, are recognized by transcriptional activators (Pdr-TFs) via their xenobiotic binding domains (XBD). Pdr-TFs bound at pleiotropic drug resistance promoter elements (PDRE) stimulate via their transactivation domains (AD) the expression of *PDR*, *MDR*, or *MFS* genes encoding plasma membrane transporters of the pleiotropic drug resistance, multidrug resistance, or major facilitator superfamilies. Lower panel: molecular mechanisms leading to antifungal drug resistance via the PDR system.

## Data Availability

Not applicable.

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
