# Peer review of "Fungal Drug Response and Antimicrobial Resistance"

_jof, 2023, doi:10.3390/jof9050565_

Round 1

Reviewer 1 Report

The article “Fungal Drug Response and Antimicrobial Resistance” authored by Trenor et al is well written. The following minor comments need to be addressed before it is accepted for publication:

1.      Reference must be cited for a few of the strong statements in the manuscript. For example, statements ending in line 39, line 75, and line 84 need to be substantiated with relevant reference

2.      It is suggested to include structures of a few antifungal drugs quoted in the manuscript so that even the chemistry readers can try correlating the structures with AMR, if any.

3. Concluding remarks must also be elaborated by giving the author’s viewpoint.

Author Response

  1. Reference must be cited for a few of the strong statements in the manuscript. For example, statements ending in line 39, line 75, and line 84 need to be substantiated with relevant reference

Response: We have now introduced the relevant references for statements in lines 39, 75 and 84.

  1. It is suggested to include structures of a few antifungal drugs quoted in the manuscript so that even the chemistry readers can try correlating the structures with AMR, if any.

Response: We have now included in Figure 1 chemical structures for representative antifungal drugs discussed throughout the manuscript.

  1. Concluding remarks must also be elaborated by giving the author’s viewpoint.

Response: We have now substantially extended the “Concluding Remarks” section by adding our viewpoint and discussing several emerging strategies to combat antifungal resistance in the future.

Reviewer 2 Report

In this review, with the title "Fungal Drug Response and Antimicrobial Resistance", the authors are providing a comprehensive overview of the serious threat to human health posed by emerging resistance to most antifungal drugs. The manuscript is well written, provides well documented informations and appears nicely up to date.

One minor comment, probably just reflecting personal taste, is that very long sentences providing a sort of list (for example the opening sentence of the introduction, or the sentence in between line 60-65) , might be easier to reed if written really as a pointed list ( a, b c and so on).

The second minor comment is that he concluding remarks are a bit short, focusing only on one promising approach (targeting the resistance mechanism themselves), while there are other approaches available that can be as well mentioned.

But overall a nicely written, clear and on point review.

Author Response

One minor comment, probably just reflecting personal taste, is that very long sentences providing a sort of list (for example the opening sentence of the introduction, or the sentence in between line 60-65) , might be easier to reed if written really as a pointed list ( a, b c and so on).

Response: The mentioned long sentences have been rephrased now as point by point lists (i, ii, iii, iv etc).

The second minor comment is that he concluding remarks are a bit short, focusing only on one promising approach (targeting the resistance mechanism themselves), while there are other approaches available that can be as well mentioned.

Response: We have now substantially extended the “Concluding Remarks” section by adding our viewpoint and discussing several emerging strategies to combat antifungal resistance in the future.